# Plumbagin, a Natural Product with Potent Anticancer Activities, Binds to and Inhibits Dihydroorotase, a Key Enzyme in Pyrimidine Biosynthesis

**DOI:** 10.3390/ijms22136861

**Published:** 2021-06-25

**Authors:** Hong-Hsiang Guan, Yen-Hua Huang, En-Shyh Lin, Chun-Jung Chen, Cheng-Yang Huang

**Affiliations:** 1Life Science Group, Scientific Research Division, National Synchrotron Radiation Research Center, Hsinchu 30076, Taiwan; d938245@oz.nthu.edu.tw; 2School of Biomedical Sciences, Chung Shan Medical University, No. 110, Sec.1, Chien-Kuo N. Rd., Taichung City 402, Taiwan; cicilovev6@gmail.com; 3Department of Beauty Science, National Taichung University of Science and Technology, No. 193, Sec.1, San-Min Rd., Taichung City 403, Taiwan; eslin7620@gmail.com; 4Department of Biotechnology and Bioindustry Sciences, National Cheng Kung University, Tainan City 701, Taiwan; 5Department of Physics, National Tsing Hua University, Hsinchu 30043, Taiwan; 6Department of Biological Science and Technology, National Chiao Tung University, Hsinchu 300193, Taiwan; 7Department of Medical Research, Chung Shan Medical University Hospital, No. 110, Sec.1, Chien-Kuo N. Rd., Taichung City 402, Taiwan

**Keywords:** anticancer, dihydroorotase, pyrimidine biosynthesis, plumbagin, crystal structure, CAD, drug design, 4T1 cell, dihydropyrimidinase, allantoinase

## Abstract

Dihydroorotase (DHOase) is the third enzyme in the de novo biosynthesis pathway for pyrimidine nucleotides, and an attractive target for potential anticancer chemotherapy. By screening plant extracts and performing GC–MS analysis, we identified and characterized that the potent anticancer drug plumbagin (PLU), isolated from the carnivorous plant *Nepenthes miranda*, was a competitive inhibitor of DHOase. We also solved the complexed crystal structure of yeast DHOase with PLU (PDB entry 7CA1), to determine the binding interactions and investigate the binding modes. Mutational and structural analyses indicated the binding of PLU to DHOase through loop-in mode, and this dynamic loop may serve as a drug target. PLU exhibited cytotoxicity on the survival, migration, and proliferation of 4T1 cells and induced apoptosis. These results provide structural insights that may facilitate the development of new inhibitors targeting DHOase, for further clinical anticancer chemotherapies.

## 1. Introduction

The de novo pyrimidine biosynthesis pathway is an attractive target for drug design [1,2,3,4,5]. In this pathway, cells generate the required pyrimidine as a building block of the DNA, for DNA replication and protein synthesis [6,7,8,9]. The pharmacological inhibition of this pathway may be used to target cancer cells, malarial parasites, and pathogens undergoing rapid growth [10,11,12,13]. The knowledge of the structure and on how these enzymes can be inhibited in this pathway is an advantage for the development of inhibitors.

Metabolism in cancer cells is usually reprogrammed in favor of the maximum anabolic synthesis of macromolecules for survival and proliferation [14]. The urea cycle, in multiple types of cancer, is commonly rewired to support anabolism [15]. When the urea cycle is dysregulated, nitrogen will be redirected and used by the multifunctional enzyme CAD (carbamoyl phosphate synthetase (CPSase)/aspartate transcarbamoylase (ATCase)/ dihydroorotase (DHOase)) to increase pyrimidine synthesis [16]. The increased expression and enzyme activity of CAD is closely linked to nucleotide synthesis and cancer cell proliferation [17,18].

DHOase catalyzes the reversible cyclization of *N*-carbamoyl aspartate (CA-asp) to dihydroorotate (DHO), for the biosynthesis of pyrimidine nucleotides (Figure 1) [12]. In mammals, DHOase is a part of a single trifunctional enzyme CAD. In most prokaryotic organisms, CPSase, ATCase, and DHOase are expressed separately and function independently [19]. In fungi, CPSase and ATCase are present in the single bifunctional protein Ura2, which is a CAD-like polypeptide that contains a defective DHOase-like domain [20]. Yeasts have a monofunctional DHOase, encoded by an independent gene, which is the protein that is crystallized in this study. Three major groups of DHOases are classified based on sequence, structural, and phylogenetic analyses. Despite evolutionary divergence, analysis of the crystal structures of DHOases from *Bacillus anthracis* (BaDHOase; the type I DHOase) [21], *Escherichia coli* (EcDHOase; the type II DHOase) [22], and the DHOase domain of human CAD (huDHOase; the type III DHOase) [23,24] reveals that an important loop, as a lid within the active site of DHOase for catalysis and substrate binding, is conserved from *E. coli* to humans. Thus, this loop in any DHOase could be the prime target for drug design.

Plant-derived herbs and drugs have been traditionally used as anti-tumor agents, alone or in combination, for many centuries and are increasingly used in modern societies. Approximately 60% of anticancer drugs on the market are derived from, or inspired by, natural products. Plumbagin (PLU), a naphthoquinone isolated from *Plumbago* plants, exerts an anticancer effect on various cell lines, at lower concentrations compared with existing chemical chemotherapeutics [25,26]. PLU exhibits an anticancer effect by inducing apoptosis, autophagic pathways, cell cycle arrest, anti-angiogenic pathways, anti-invasion pathways, and anti-metastasis pathways [26]. In animal models, PLU-treated mice showed a significant reduction in tumor growth, and no side effects [27]. PLU is a vitamin K3 analog. PLU has been studied for more than 40 years; however, no PLU-complexed protein structure is available to date. This complex structure is needed as a molecular basis to formulate any inhibition model.

In contrast to normal cells, cancer cells need to constitutively express and activate DHOase for the biosynthesis of pyrimidines, for rapid cell growth and proliferation [15,16]. The potent DHOase inhibitor may have strong cytotoxicity against cancer cells. The structures of DHOase complexed with DHO analogs, such as 5-fluorouracil (5-FU) [28], 5-aminouracil (5-AU) [28], and 5-fluoroorotate (5-FOA) [22,24], were solved. In the present study, the potent anticancer drug PLU was found to be a novel and non-DHO analog inhibitor of DHOase. We also solved the PLU-complexed crystal structure of DHOase to determine the binding interactions and investigate the binding mode. This study aimed to find a new inhibitor of DHOase and investigate the structure–inhibition relationships for further anticancer drug development.

## 2. Results

### 2.1. Inhibition of DHOase by Using Substrate Analogs

Considering that a substrate analog for any enzyme is usually a potential inhibitor, we initially tested compounds that are similar to DHO, but have different ring structures (Figure 2A–C), such as 2,7-dioxo-1,3-diazepane-4-carboxylic acid (DDCA; DHO analog, but with a seven-membered ring), 2-imino-4-oxo-1,3-thiazinane-6-carboxylic acid (ITCA; DHO analog, but with S-containing a six-membered ring), and 5-hydantoinacetic acid (5-HAA; DHO analog, but with a five-membered ring), as inhibitors of huDHOase. In the enzyme assay, DDCA (200 μM; Figure 2A) and 5-HAA (200 μM; Figure 2C) did not inhibit huDHOase. Meanwhile, the S-containing six-membered ring compound ITCA (200 μM; Figure 2B) reduced the activity of huDHOase by 12%. Thus, ITCA was identified as a new inhibitor, but only slightly inhibited the activity of huDHOase.

We attempted to test DDCA, ITCA, and 5-HAA as substrates of huDHOase. After incubation with huDHOase for 2 h, DDCA (1 mM), ITCA (1 mM), and 5-HAA (10 mM) were not hydrolyzed; therefore, these compounds with different ring sizes were determined as nonsubstrates of huDHOase.

Further, 5-FOA (Figure 2D) is a potent inhibitor of DHOase from the malaria parasite *Plasmodium falciparum*, that is, it exhibited 50% growth inhibition of parasites at a concentration of 6.0 nM [29]. We used 5-FOA as an inhibitor of huDHOase. In contrast to the significant effect on *P. falciparum* DHOase, 200 μM 5-FOA in the standard assay decreased the activity of huDHOase by only 26%.

We evaluated whether the inhibitors of the metalloenzyme glutaminyl cyclase [30], namely, N-ω-acetylhistamine (Figure 2E), 3,5-diamino-1,2,4-triazole (Figure 2F), 3-amino-1,2,4-triazole (Figure 2G), and 5-methylthio-1H-tetrazole (Figure 2H), were potential inhibitors of huDHOase. None of these compounds decreased the activity of huDHOase. Acetohydroxamate (Figure 2I), an inhibitor of allantoinase (ALLase), was not an inhibitor of huDHOase. The anti-HIV drug AZT [31], and the anticancer drugs 5-FU (Figure 2J) and 5-AU (Figure 2K) also did not inhibit the activity of huDHOase. Malate (200 μM; Figure 2L) decreased the activity of huDHOase by 6% (Figure 2M).

### 2.2. Inhibition of DHOase by Plant Extracts

Given the poor inhibition ability of substrate analogs, we screened for new DHOase inhibitor(s) from natural products. We prepared different extracts from *Sarracenia purpurea*, *Nepenthes Miranda*, *Prunus mume*, *Cyphomandra betacea* Sendt, and *Passiflora edulis*, to determine the possible inhibitory effect on DHOase. The *S. purpurea*, *P. mume*, *C. betacea* Sendt, and *P. edulis* extracts did not affect the huDHOase activity. However, the *N. miranda* extracts (i.e., leaves, stem, and pitcher, obtained through 100% acetone) inhibited huDHOase. Thus, certain compound(s) abundant in the acetone fraction of the *N. miranda* extract could be a potential DHOase inhibitor.

The chemical composition of the leaf extract from *N. miranda* was analyzed through GC–MS, given its relative abundance and availability. The top content (28.52%) in the leaf extract of *N. miranda* was PLU. PLU has a wide range of biological effects, including cytotoxicity against cancer cells in vitro and in vivo [26]. Based on the results, we purchased and used PLU as an inhibitor of the purified huDHOase, the eukaryotic *Saccharomyces cerevisiae* DHOase (ScDHOase), *Salmonella enterica* serovar Typhimurium LT2 DHOase (StDHOase), and *Klebsiella pneumonia* DHOase (KpDHOase). PLU (200 μM) decreased the activities of StDHOase, KpDHOase, ScDHOase, and huDHOase by 71%, 69%, 66%, and 30%, respectively (Figure 3). Thus, PLU exhibited a greater inhibitory effect on huDHOase than the substrate analog 5-FOA.

### 2.3. Crystal Structure of ScDHOase Complexed with PLU

After the initial crystallization screening for DHOases, only the PLU–ScDHOase complex formed crystals. The crystal of the PLU–huDHOase complex cannot be obtained for structure determination; as such, ScDHOase, rather than huDHOase, was used as the model for studying the structure–inhibition relationship.

The complexed crystal structure of ScDHOase with PLU should be solved to determine where the binding occurred and investigate the binding modes. We attempted to crystallize the PLU–ScDHOase complex by crystallization screening, but no crystal was formed. We found that ScDHOase formed crystals grown in the presence of malate [28]; therefore, we incubated PLU with the crystal of the malate–ScDHOase complex. In contrast to the case of forming the 5-FU–ScDHOase complex [28], the crystals incubated with PLU, to form the PLU–ScDHOase complex, were extremely unstable and disintegrated shortly. It is possible that some regions in ScDHOase became dynamic due to the entry of PLU into the active site.

The crystals of the PLU–ScDHOase complex belonged to space group P2_1_ (Table 1). Four monomers of ScDHOase were found in the asymmetric unit (Figure 4A). The Lys residue (K98) remained carbamylated, regardless of PLU binding. The binding by PLU did not change the metal content of ScDHOase, that is, the dimetal center (Znα/Znβ) in ScDHOase, containing four His (i.e., H14, H16, H137, and H180), one Asp (i.e., D258), and one carbamylated Lys (i.e., Kcx98), was still self-assembled. The structure also revealed a flexible loop that extended toward the active site when PLU was bound (Figure 4B).

PLU was found in the active site (Figure 5 and Appendix A), only in subunits B (Figure 5B) and D (Figure 5D). Malate was occupied in subunits A (Figure 5A) and C (Figure 5C). These PLUs were bound by ScDHOase in different modes. His16, Arg18, Asn43, and Lys230 in subunit B were involved in PLU binding (Figure 5B), whereas Thr105 and Asp258 in subunit D interacted with PLU (Figure 5D). Despite the different binding poses, the flexible loops in subunits B and D had a similar loop-in conformation (Figure 4B). Stably binding the PLU molecule within the active site may function similarly to firm stuffing, to prevent the entry of substrate CA-asp from the dynamic loop, as the molecular size of PLU is larger than that of CA-asp (see below for discussion).

### 2.4. Identification of PLU as a Competitive Inhibitor of DHOase

For the first time, the potent anticancer drug PLU, a bicyclic compound without similarity to DHO, was identified as an inhibitor of DHOase. PLU was included in the standard assay for the analysis of enzyme activity under different substrate concentrations, to determine the inhibition type. The Lineweaver–Burk plot, with lines crossing the y-axis at a similar point, indicated competitive inhibition on ScDHOase (Figure 6). The *V*_max_ and *K*_m_ values of ScDHOase were 64.1 ± 2.8 μmol/mg/min and 0.115 ± 0.024 mM, respectively, in the presence of PLU, and 69.9 ± 1.0 μmol/mg/min and 0.029 ± 0.003 mM, respectively, without PLU. The *K*_m_ value increased by four-fold, whereas the *V*_max_ value was only slightly affected. The maximum rate was unaffected, which reflects the fact that the inhibitor and substrate are competing for the same active site. On the basis of the results from the kinetic analysis, PLU was identified as a competitive inhibitor and could compete with DHO for the active sites of ScDHOase.

### 2.5. Mutational Analysis of Residues within the Active Site

ScDHOase has not been analyzed through mutagenesis to date. We investigated whether the active site residues in ScDHOase are critical (Table 2). A comparison of the crystal structures of ScDHOase, huDHOase, and EcDHOase indicated that His14, His16, Kcx98, His137, His180, and Asp258 in ScDHOase might be crucial for the assembly of the binuclear metal center within the active site. Arg18, Asn43, and His262 in ScDHOase might be involved in substrate binding. Thr105 and Thr106, which are important residues on the dynamic loop, might be responsible for the substrate entrance and the product release of ScDHOase. The alignment consensus of 1694 uniquely sequenced DHOase homologs, by ConSurf, revealed that these positions were highly conserved for catalysis (Appendix A). The importance of these residues was probed using site-directed mutagenesis, in which alanine substitution was constructed and analyzed (Appendix A). As shown in Table 2, the catalytic activities of the mutant ScDHOases were severely impaired. H14A, H16A, R18A, N43A, K98A, H137A, H180A, D258A, and H262A were inactive. The D258E mutant protein was found to be active, but its activity was only approximately 0.02% compared with that of the wild-type ScDHOase. The activities of T105A and T106A were less than 1% of the wild-type ScDHOase, suggesting their importance in catalysis. The structure-based mutational analysis indicated that the active site residues in ScDHOase were crucial, similar to those in huDHOase.

### 2.6. Binding Specificities of ScDHOase

The clinical anticancer drug 5-FU can bind to [28], but cannot inhibit DHOase (Figure 2). DDCA, a DHO analog with a seven-membered ring, was neither a substrate nor an inhibitor of ScDHOase (Figure 2). Whether this DHO analog (DDCA) could enter the active site of ScDHOase, similarly to 5-FU, remains unknown. We also investigated which compound could be a ligand of ScDHOase. The *K*_d_ of ScDHOase was determined through fluorescence quenching, to confirm the strength of the interaction of ScDHOase with these compounds. Quenching refers to the complex formation process that decreases the fluorescence intensity of the protein. ScDHOase displayed strong intrinsic fluorescence, with a peak wavelength of 324 nm, when excited at 280 nm. When DDCA (200 μM) was added into the ScDHOase solution, the intrinsic fluorescence was only quenched by 3.0%, suggesting low interactions (Appendix A). Thus, DDCA might not effectively bind to ScDHOase, which might explain why DDCA was neither a substrate nor an inhibitor of ScDHOase, possibly due to the ring size (seven-membered ring) of DDCA. We also determined the fluorescence quenching of ScDHOase by 200 μM ITCA, 5-HAA, 5-FOA, N-ω-acetylhistamine, 3,5-diamino-1,2,4-triazole, 3-amino-1,2,4-triazole, 5-methylthio-1H-tetrazole, acetohydroxamate, malate, PLU, and orotic acid, with values of 15.6%, 1.5%, 78.2%, 2.8%, 3.9%, 2.8%, 2.3%, 2.5%, 10.9%, 87.7%, and 65.9%, respectively. The quenching of ScDHOase by some of these compounds was less than 20%, which was too low to determine the *K*_d_ values. Of these compounds, 5-FOA (78.2%), PLU (87.7%), and orotic acid (65.9%) effectively bound to ScDHOase.

Based on the quenching result, PLU might be the best of the aforementioned compounds for the binding of ScDHOase. When different concentrations of PLU were individually titrated into the ScDHOase solution, the intrinsic fluorescence was progressively quenched (Figure 7A). PLU resulted in a blue shift (~6.5 nm; *λ*_max_ was from 324.0 to 317.5 nm) in the ScDHOase emission wavelength (*λ*_em_), whereas 5-FU and 5-AU [28] resulted in a red shift (*λ*_max_ was from 324 to 335.5 nm). This result for the *λ*_max_ shift indicated that PLU interacted and formed a stable complex with ScDHOase. The *K*_d_ value of ScDHOase bound to PLU was 64.8 ± 1.6 μM, as determined through the titration curve (Table 3).

To confirm whether PLU can bind to other DHOases, we determined the *K*_d_ values of PLU for huDHOase (Figure 7B) and StDHOase (Figure 7C), through fluorescence quenching. Further, huDHOase and StDHOase individually displayed strong intrinsic fluorescence, with peak wavelengths of 340 and 330.5 nm, when excited at 280 nm. In the titration curves, the *K*_d_ values of PLU for huDHOase and StDHOase were 150.9 ± 4.1 and 80.3 ± 2.5 μM, respectively.

The DHO analog orotic acid was neither a substrate nor an inhibitor of ScDHOase. The *K*_d_ value of orotic acid for ScDHOase was also determined through the titration curve, as 198.3 ± 1.2 μM (Appendix A). The abilities of 5-FU [28] and orotic acid to bind to ScDHOase were approximately equal.

The decrease in the intrinsic fluorescence of DHOase was measured with a spectrofluorometer (Hitachi F-2700; Hitachi High-Technologies, Japan). The *K*_d_ was obtained using the following equation: Δ*F* = Δ*F*_max_ − *K*_d_(Δ*F*/[PLU]).

### 2.7. Structure-Based Binding Analysis

Our complex structure of ScDHOase with PLU showed that all the binding processes occurred through the loop-in mode (Figure 4B), that is, the dynamic loop of ScDHOase might be involved in the binding. Two important residues in the binding were Arg18, which is responsible for ligand binding, and Thr106, which is responsible for initiating the movement of the flexible loop (loop-in mode) in the catalytic cycle of DHOase. The two residues were conserved and crucial for catalysis (Appendix A and Table 2). The binding abilities of the R18A and T106A mutants were analyzed through fluorescence quenching, and compared with those of the wild-type ScDHOase. ScDHOase–R18A (Figure 7D) and ScDHOase–T106A (Figure 7E) individually displayed strong intrinsic fluorescence, with peak wavelengths of 324.5 and 328 nm, when excited at 280 nm. The *K*_d_ values of the two mutant proteins, determined using the titration curves (Figure 7F) for the binding of PLU, were all decreased (Table 3). However, the binding abilities of the R18A and T106A mutants for PLU were not similar. For binding to PLU, the *K*_d_ values of R18A and T106A were 181 ± 5.9 and 70.8 ± 2.6 μM, respectively. Compared with that of the wild-type ScDHOase (64.8 ± 1.6 μM), T106 was not a critical residue for PLU binding.

### 2.8. Anticancer Activity of PLU against 4T1 Cells

PLU is a potent anticancer agent that has minimal side effects and less toxicity for normal cells [32] compared with other existing chemotherapeutics [26]. In this study, we identified that PLU inhibited DHOase, and might, therefore, suppress the biosynthesis of pyrimidines in cancer cells. Whether PLU can suppress the growth of 4T1 cells is undetermined. The 4T1 mammary carcinoma is a transplantable breast cancer cell line that is highly tumorigenic and invasive [33]. In contrast to most tumor models, 4T1 cells can spontaneously metastasize from the primary tumor in the mammary gland to multiple distant sites, including the lymph nodes, blood, liver, lung, brain, and bone [34,35]. Thus, the cytotoxic effect of PLU on the survival, migration, and proliferation of 4T1 mammary carcinoma cells was investigated (Figure 8). We found that 10 μM PLU was sufficient to suppress 4T1 cell activities. Incubation with 10 μM PLU caused significant cell deaths, by 100% (Figure 8A), and inhibited the cell migration (Figure 8B) by 97%. The Hoechst staining showed that 10 μM PLU induced apoptosis (Figure 8C), with DNA fragmentation in 4T1 cells (100%). The clonogenic formation assay (Figure 8D) revealed that the pretreatment with 10 μM PLU significantly suppressed the proliferation and the colony formation of 4T1 cells, by 98%. Thus, 10 μM PLU was effective for anti-4T1 cancer cells (Figure 8E).

### 2.9. PLU Inhibited Bacterial Growth

Similar to cancer cells, bacterial cells are rapidly reproducible, and their growth might be suppressed by PLU. To test whether PLU could inhibit bacterial growth, we grew *Klebsiella pneumoniae* and *S. enterica* serovar Typhimurium LT2 cells to 0.5 OD_600_ at 37 °C, and then added PLU into the medium. Figure 8F shows that 100 μM PLU significantly inhibited the growth of the two pathogens. Thus, PLU was also effective for antipathogen chemotherapies. PLU may inhibit pyrimidine biosynthesis, resulting in antibacterial activities.

### 2.10. PLU Could Act with 5-FU for Anti-4T1 Cancer Cells

The FDA-approved clinical drug 5-FU has a remarkable therapeutic effect against different cancers [9]. We investigated whether PLU could act with 5-FU against 4T1 cancer cells (Figure 9A). The cytotoxicity experiments showed that 5-FU acting with PLU had synergistic anti-4T1 cancer cell effects. The usage of PLU (2 μM) and 5-FU (5 μM) led to 8% and 27% cell mortality, respectively. The cytotoxic effect was significantly enhanced, and the cell mortality was 59% (Figure 9B). The cotreatment of PLU with 5-FU on 4T1 cells also had greater effects on apoptotic rate (Figure 9C), colony formation (Figure 9D), and cell migration (Figure 9E). Thus, the usage of PLU with 5-FU resulted in increased cytotoxicity against 4T1 cells (Appendix A).

## 3. Discussion

### 3.1. Identification of DHOase Inhibited by PLU

The development of clinically useful small-molecule drugs, and the identification of new key targets have been a seminal event in the field of anticancer chemotherapies. Nucleotide synthesis is essential to the maintenance of homeostasis and the proliferation of cancer cells [5], and thus should be a prime target in the development of anticancer drugs [1,5]. DHOase is required for pyrimidine biosynthesis; as such, blocking the activity of this enzyme would be detrimental to cancer cell survival. DHOase activity is not constitutively necessary for normal cells; hence, DHOase inhibitors are potentially safe for further medical development.

To develop new DHOase inhibitors, we prepared different extracts from *S. purpurea*, *N. miranda*, *P. mume*, *C. betacea* Sendt, and *P. edulis*, and determined their possible inhibitory effect on DHOase. The results of the GC–MS analysis showed that PLU, which was isolated from the leaf extract of the carnivorous plant *N. miranda*, was found to be a novel inhibitor of DHOase (Figure 3). PLU, a naphthoquinone-derived phytochemical, does not resemble DHO, but can inhibit DHOase. PLU has a wide range of biological activities and pharmaceutical relevance in cancer therapy [26]. The selective cytotoxicity of PLU is due to its ability to regulate multiple cancer signaling pathways, such as FZD–Wnt, EGFR, and NF-κB signaling [26,32]. PLU can also increase intracellular ROS and activate p53, to induce apoptosis and reduce tumor growth and weight [32]. Possibly, PLU can regulate cellular nucleotide synthesis to suppress cancer cell growth and survival (Figure 8), through the inhibition of DHOase (Figure 3).

### 3.2. PLU as a Dirty Drug for Multiple Targets

PLU exerts cytotoxicity by targeting several molecular mechanisms, including apoptosis and autophagic pathways, cell cycle arrest, antiangiogenic pathways, anti-invasion pathways, and antimetastasis pathways [26,36,37]. In addition to DHOase, PLU inhibits DHPase activity [38], with an IC_50_ value of 58 μM [39]. Given that DHPase is an important enzyme in pyrimidine metabolism, PLU may be a competent dirty drug for multiple targets. PLU could regulate pyrimidine metabolism and the resulting cellular signaling by targeting DHOase and DHPase simultaneously.

From a biochemical point of view, DHOase [10,40] is a member of the cyclic amidohydrolase family [38,41], which also includes DHPase [42,43,44,45,46], ALLase [47,48,49], hydantoinase (HYDase) [50,51], and imidase [52,53,54]. These metal-dependent enzymes catalyze the hydrolysis of the cyclic amide bond of each substrate, in either five- or six-membered rings, in the metabolism of purines and pyrimidines. Almost all enzymes contain the binuclear metal center that consists of four His, one Asp, and one post-translational carbamylated Lys. Although these cyclic amidohydrolases may use a similar active site and mechanism for catalysis [38], no substrate overlapping was observed for DHOase, DHPase, and ALLase [55]. The post-carbamylated Lys is needed for the enzyme activity and the self-assembly of the binuclear metal center [47,50,56]. PLU may inhibit ALLase, HYDase, and imidase due to their similar active sites. However, this speculation should be further elucidated biochemically and structurally.

### 3.3. Loop-in Binding Mode of PLU and Malate

The first reported structure of DHOase is that from *E. coli* [40]. In the structure of a dimeric EcDHOase, the substrate CA-asp and the product DHO are found at different active sites. The structure also reveals a flexible loop (Figure 10A) that extends toward the active site when CA-asp is bound (the loop-in mode), or moves away from the active site, facilitating the release of the product DHO (the loop-out mode). In addition, 5-FOA, a product-like inhibitor, binds to the active site of EcDHOase in a similar manner to DHO, via the loop-out binding mode, that is, the loop does not interact with the ligand or with the rest of the active site [22]. However, our complexed structures of ScDHOase with 5-AU [28], 5-FU [28], malate (Figure 5), and PLU (Figure 5) revealed that the binding occurred through the loop-in mode. In addition, the complexed structures of ScDHOase with the weaker inhibitor malate, determined at pH 6.0 to 9.0, revealed preference for the loop-in binding mode, regardless of pH [28]. Whether these different binding modes are species- or crystallography-dependent should be elucidated experimentally for drug optimization.

### 3.4. Dynamic Loop as a Part of the Catalytic Cycle in DHOase and DHPase and as a Drug Target

Based on the crystal structures of HYDase [57], DHPase [46], DHOase [40,58], and ALLase [49], the chemical mechanism of the binuclear metal center-containing cyclic amidohydrolases potentially has the following three main steps: (I) the hydrolytic water molecule must be activated for nucleophilic attack; (II) the amide bond of the substrate must be made electrophilic by the polarization of the carbonyl O bond; (III) the leaving group N must be protonated as the C–N bond is cleaved. The flexible loop [22] in EcDHOase is crucial for stabilizing the transition state and catalysis, supporting that the movement of this loop is a part of the catalytic cycle, and also one of the main steps (Figure 10A). Our mutational analysis indicated the importance of the two Thr residues (Thr105 and Thr106) on the loop in ScDHOase (Table 2). This case is similar to that of DHPase. The conserved Tyr residue, located within a dynamic loop in DHPase, plays an essential role in the stabilization of the tetrahedral transition state during the hydrolysis of the substrate, collapse of the transition state, formation of a product, and release of the product [38,59]. Thus, the dynamic loop in DHOase and DHPase could be a suitable drug target for inhibiting pyrimidine metabolism [10,39].

### 3.5. Active Site Distorted by PLU

Analysis of the 5-FU- and 5-AU-complexed structures of ScDHOase indicated loop-in binding mode [28]; however, they were not inhibitors (Figure 2). Thus, preventing the dynamic loop from movement and stabilizing the transition state might not be sufficient for the strong inhibition of DHOase. For example, malate can bind to DHOase via the loop-in mode, but only with weak inhibition. Comparatively, PLU is a bicyclic compound that can inhibit DHOases, whereas smaller monocyclic compounds, namely, 5-FU and 5-AU, could bind, but could not inhibit, any DHOase. Based on structural evidence, we found that the distances between the metal-binding residue Asp258 and Znα, in subunits B and D, were different. The distances of Asp258–Znα in subunits D and B were 1.98 (Figure 5D) and 5.12 Å (Figure 5B), respectively. The distance of Asp258–Znα in subunits B was 5.12 Å, which suggests that it has no role in the interaction with Znα. In addition, Asp258 in subunit D was shifted by a distance of 2.9 Å and an angle of 30.8° for hydrogen bonding to PLU (Figure 5D and Figure 10B). Given that Asp258 (Asp250 in EcDHOase) plays dual crucial roles in the catalysis, namely, as the general base for the nucleophilic attack [60] and the metal binding of DHOase [61,62], the difference in this active site architecture would significantly affect the catalysis behavior. Hence, we speculated that the deviation of Asp258, caused by the binding of PLU, would distort the active site of ScDHOase, thereby inactivating ScDHOase function (Figure 10B). However, the complex structure could not be obtained in a high resolution. Further structural study is still needed to investigate the binding mode of PLU.

## 4. Materials and Methods

### 4.1. Protein Expression and Purification

The construction of StDHOase [61], KpDHOase [62], huDHOase [61], and ScDHOase [61,63] expression plasmids were reported previously. The recombinant protein was purified using the protocol described previously for SSB-like proteins [64,65,66,67,68]. Briefly, *E. coli* BL21(DE3) cells were transformed with the expression vector, and the overexpression of the expression plasmids was induced by incubating with 1 mM isopropyl thiogalactopyranoside. The protein was purified from the soluble supernatant by using Ni^2+^-affinity chromatography (HiTrap HP; GE Healthcare Bio-Sciences), eluted with buffer A (20 mM Tris–HCl, 250 mM imidazole, and 0.5 M NaCl, pH 7.9), and dialyzed against a dialysis buffer (20 mM Tris–HCl and 0.1 M NaCl, pH 7.9). The protein purity remained at >97% as determined using SDS–PAGE (Mini-PROTEAN Tetra System; Bio-Rad, CA, USA).

### 4.2. Site-Directed Mutagenesis

The ScDHOase mutants were generated according to the QuikChange site-directed mutagenesis kit protocol (Stratagene; LaJolla, CA, USA), by using the wild-type plasmid pET21b–ScDHOase as a template. The presence of the mutation was verified by DNA sequencing in each construct. The recombinant mutant proteins were purified using the protocol for the wild-type ScDHOase by Ni^2+^-affinity chromatography.

### 4.3. Crystallization Experiments

Before crystallization, the purified ScDHOase was concentrated to 11 mg/mL. The crystals of ScDHOase were grown at room temperature through hanging drop vapor diffusion in 16% PEG 4000 and 100 mM imidazole–malate. The crystals of the malate–ScDHOase complex reached full size in 9–12 days. PLU (200 μM) was incubated with the crystal of the malate–ScDHOase complex for 30 min. The crystals were transferred from a crystallization drop into the cryoprotectant solution (2 μL) with precipitant solution containing glycerol (25–30%) for a few seconds, mounted on a synthetic nylon loop (0.1–0.2 mm, Hampton Research), and flash cooled in liquid N_2_. The crystals of ScDHOase were validated in the beamline 15A1 of the National Synchrotron Radiation Research Center (NSRRC; Hinchu, Taiwan).

### 4.4. X-ray Diffraction Data and Structure Determination

The native and the Zn-anomalous data were collected at beamline BL44XU at SPring-8 (Harima, Japan) with an MX300-HE CCD detector and at beamline TPS 05A at the NSRRC (Hsinchu, Taiwan) with an MX300-HS CCD detector. Data sets were indexed, integrated, and scaled by HKL-2000 [69] and XDS [70]. The initial phase, density modification, and model building were performed using the AutoSol program [71] in the PHENIX. The iterative model building and structure refinement were performed using Refmac in the CCP4 software suite [72] and phenix.refine in the PHENIX software suite [73]. The initial phase of ScDHOase complexed with PLU was determined through the molecular replacement software Phaser MR [74] by using the monomeric ScDHOase derived from ScDHOase–malate complex [30] as the search model. The correctness of the stereochemistry of the models was verified using MolProbity [75]. Atomic coordinates and related structure factors were deposited in the PDB with accession code 7CA1.

### 4.5. Plant Materials and Extract Preparations

Different plant extracts were prepared from the leaves of *S. purpurea*, *N. miranda*, *P. mume*, *C. betacea* Sendt, and *P. edulis* to test the possible inhibitory effect on ScDHOase. Leaves were collected, dried, cut into small pieces, and pulverized into powder. Extractions were carried out by placing 1 g plant powder into a 250 mL conical flask. The flask was added with 100 mL solvents (i.e., methanol, ethanol, acetone, or distilled water) and shaken on an orbital shaker for 5 h. PLU was found from the extract of *N. miranda* obtained by 100% acetone.

### 4.6. Gas Chromatography–Mass Spectrometry (GC–MS)

The chemical composition of the plant extract was analyzed through GC–MS. Briefly, the filtered sample was analyzed using the Thermo Scientific TRACE 1300 gas chromatograph with the Thermo Scientific ISQ single-quadrupole mass spectrometer system. The column used was the Rxi-5ms (30 m × 0.25 mm i.d. × 0.25 μm film). The temperature of the injection port was 300 °C, and the flow rate of the He was 1 mL·min^−1^. The compounds discharged from the column were detected using a quadrupole mass detector. The ions were generated using the electron ionization method. The relative mass fraction of each chemical component was determined using the peak area normalization method. Compounds were identified by matching the generated spectra with the NIST 2011 and the Wiley 10th edition mass spectral libraries.

### 4.7. Enzyme Assay

A rapid spectrophotometric assay was used to determine the activity of DHOase [47,55]. Briefly, the hydrolysis of DHO was measured at 25 °C as the decrease in absorbance at 230 nm. The purified DHOase was added to a 2 mL solution containing 0.5 mM DHO and 100 mM Tris–HCl at pH 8.0 to start the reaction. The extinction coefficient of DHO was 0.92 mM^−1^·cm^−1^ at 230 nm. The hydrolysis of DHO was monitored using a UV/Vis spectrophotometer (Hitachi U 3300; Hitachi High-Technologies, Tokyo, Japan). A unit of activity was defined as the amount of enzyme catalyzing the hydrolysis of 1 μmol DHO/min, and the specific activity was expressed in terms of units of activity per mg of enzyme. The kinetic parameters *K*_m_ and *V*_max_ were determined from a nonlinear plot by fitting the hydrolyzing rate from individual experiments to the Michaelis–Menten equation (Enzyme Kinetics module of Sigma-Plot; Systat Software, Chicago, IL, USA).

### 4.8. Determination of the Dissociation Constant (K_d_)

The *K*_d_ value of the purified DHOase was determined using the fluorescence quenching method as previously described for DHOase [30,55], DHPase [35,44,76], and DnaB helicase [77,78]. Briefly, an aliquot of the compound was added into the solution containing DHOase (1 μM) and 50 mM HEPES at pH 7.0. The decrease in the intrinsic fluorescence of DHOase was measured at 324 nm upon excitation at 280 nm and 25 °C with a spectrofluorometer (Hitachi F-2700; Hitachi High-Technologies, Japan). The *K*_d_ was obtained using the following equation: Δ*F* = Δ*F*_max_ − *K*_d_(Δ*F*/[PLU]).

### 4.9. Trypan Blue Cytotoxicity Assay

The trypan blue cytotoxicity assay was performed to assess the cell death and study whether PLU can exhibit cytotoxicity on the 4T1 carcinoma cell survival. The 4T1 cells (1 × 10^4^) were incubated with PLU in a 100 μL volume. After 24 h, the anticancer potentiality was estimated [79].

### 4.10. Wound-Healing Assay

The wound-healing assay was performed to study whether PLU can inhibit 4T1 cell migration [80]. Briefly, 4T1 cells were seeded in 24-well plates, incubated in serum-reduced medium for 6 h, wounded in a line across the well with a 200 μL pipette tip, and washed twice with the serum-reduced medium. After different treatments, cells were incubated for 24 h to allow migration.

### 4.11. Clonogenic Formation Assay

A clonogenic formation assay was used to assess the 4T1 cell growth to study whether the PLU can inhibit 4T1 cell proliferation [81]. Briefly, 4T1 cells were seeded at a density of 10^3^ cells per well into 6-well plates and incubated overnight for attachment. After different treatments, plates were incubated for 5–7 days to allow clonogenic growth. After washing with PBS, colonies were fixed with methanol and stained with 0.5% crystal violet for 20 min, and the number of colonies was counted under a light microscope.

### 4.12. Chromatin Condensation Assay

The apoptosis in 4T1 cells was assayed with Hoechst 33342 staining [82]. The 4T1 cells were seeded in 6-well plates at a density of 5 × 10^5^ cells per well in a volume of 100 mL of culture medium. Cells were allowed to adhere for 16 h. After different treatments, cells were incubated for an additional 24 h, washed with PBS, stained with the Hoechst dye (1 μg·mL^−1^) in the dark at RT for 10 min, and imaged using an inverted fluorescence microscope (Axiovert 200 M; Zeiss Axioplam, Oberkochen, Germany) at excitation and emission wavelengths (*λ*_em_) of 360 and 460 nm, respectively. The apoptotic index was calculated as follows: apoptotic index = apoptotic cell number/(apoptotic cell number + nonapoptotic cell number).

## 5. Conclusions

By screening plant extracts, we identified and characterized that the potent anticancer agent PLU is a competitive inhibitor of DHOase. PLU can inhibit DHOase and may, therefore, suppress pyrimidine biosynthesis. The usage of PLU with 5-FU resulted in increased cytotoxicity against 4T1 cells. We also determined the complexed crystal structure, which was the first structure for PLU complexed with a protein in PDB.

## Figures and Tables

**Figure 1 ijms-22-06861-f001:**
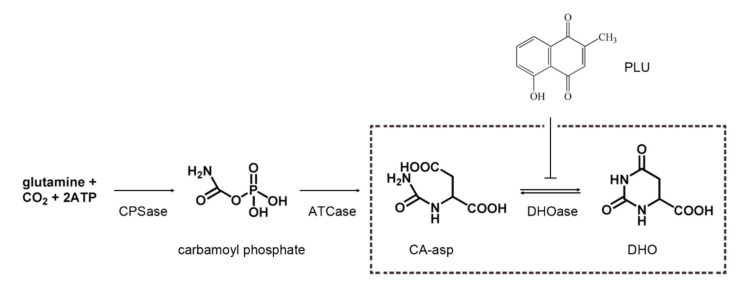
The gene products for the first three reactions of pyrimidine biosynthesis. DHOase catalyzes the reversible cyclization of CA-asp to DHO. In the present study, the potent anticancer drug plumbagin (PLU) was found to be a novel and non-DHO analog inhibitor of DHOase.

**Figure 2 ijms-22-06861-f002:**
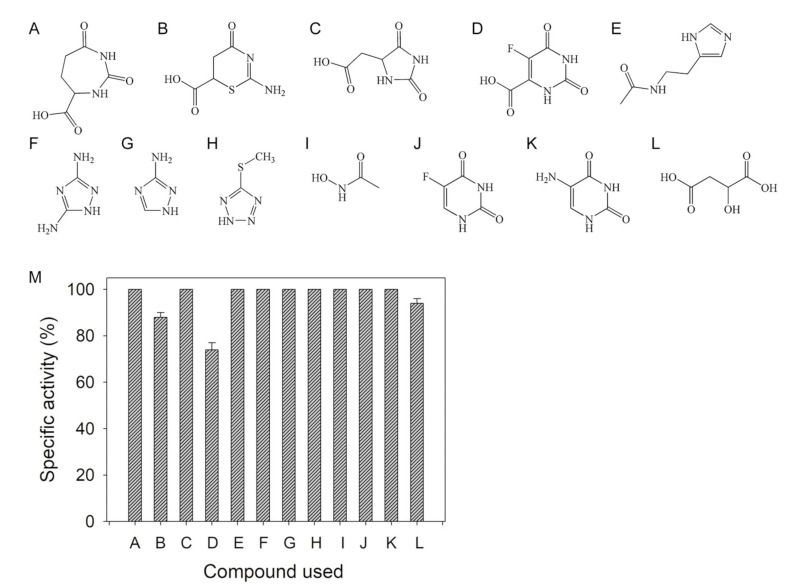
Inhibition of DHOase by substrate analogs. Molecular structure of (**A**) 2,7-dioxo-1,3-diazepane-4-carboxylic acid, (**B**) 2-imino-4-oxo-1,3-thiazinane-6-carboxylic acid, (**C**) 5-hydantoinacetic acid, (**D**) 5-FOA, (**E**) N-ω-acetylhistamine, (**F**) 3,5-diamino-1,2,4-triazole, (**G**) 3-amino-1,2,4-triazole, (**H**) 5-methylthio-1H-tetrazole, (**I**) acetohydroxamate, (**J**) 5-FU, (**K**) 5-AU, and (**L**) malate. (**M**) Effect of the substrate analog (200 μM) on the activity of huDHOase. Concentration of 200 μM 2-imino-4-oxo-1,3-thiazinane-6-carboxylic acid, 5-FOA, and malate in the standard assay decreased the activity of huDHOase by 12%, 26%, and 6%, respectively.

**Figure 3 ijms-22-06861-f003:**
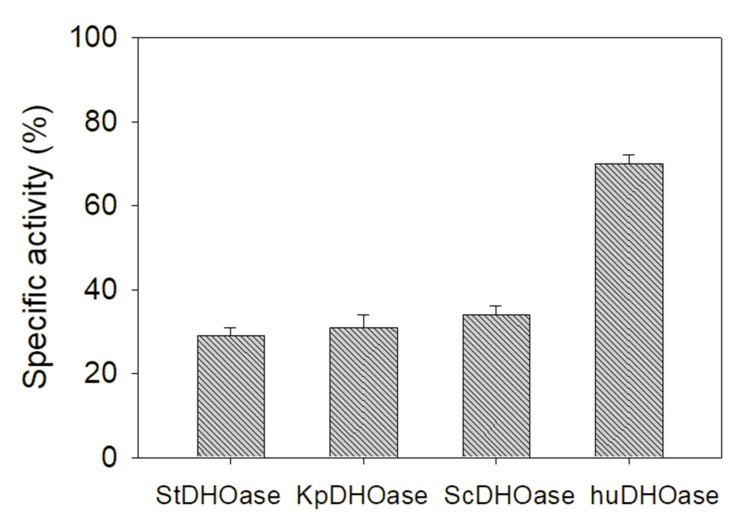
Inhibition specificity of PLU. PLU (200 μM) decreased the activities by 71%, 69%, 66%, and 30%, respectively.

**Figure 4 ijms-22-06861-f004:**
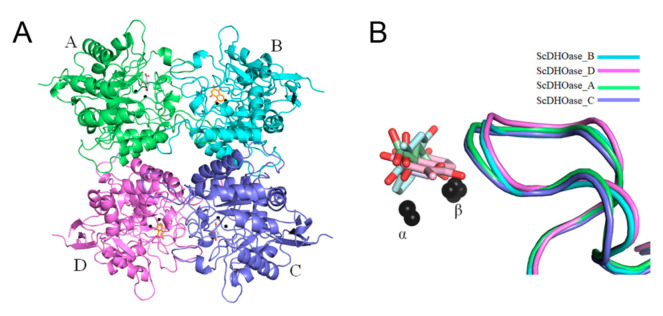
Structure of ScDHOase complexed with PLU. (**A**) Ribbon diagram of the PLU–ScDHOase complex tetramer. Each monomer is color-coded. Two zinc ions in the active site are presented as black spheres. PLU (bright orange) was found in the active site only in subunits B and D. Malate (slate) was occupied in subunits A and C. (**B**) Structural comparison of the active sites. The loops (amino acid residues 99–115) in different monomers of ScDHOase are color-coded. The superimposed structures of the PLU-bound states and the malate-bound states revealed that the flexible loop is in similar loop-in conformation.

**Figure 5 ijms-22-06861-f005:**
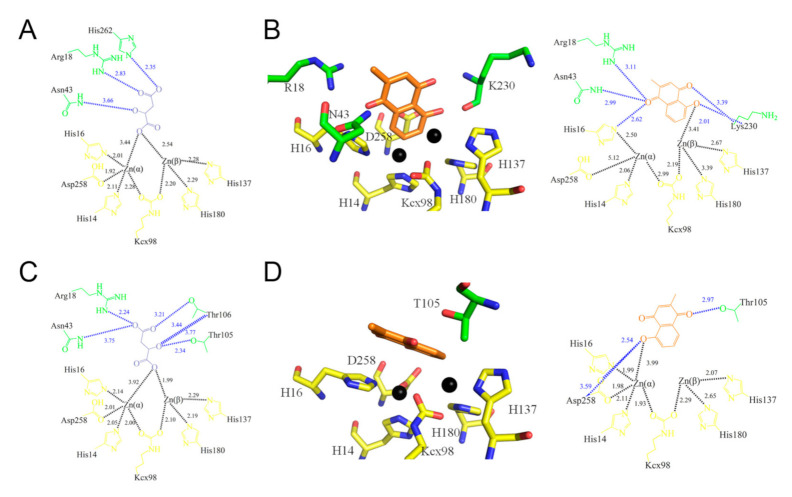
PLU and malate binding modes. (**A**) The active site of subunit A with malate (in dark blue). Arg18, Asn43, and His262 were involved in malate binding. Residues required for metal binding are colored in yellow. (**B**) The active site of subunit B with PLU (in orange). His16, Arg18, Asn43, and Lys230 were involved in PLU binding. Zn atoms are shown in black spheres. (**C**) The active site of subunit C with malate. Arg18, Asn43, Thr105, and Thr106 were involved in malate binding. (**D**) The active site of subunit D with PLU. Thr105 and Asp258 were involved in PLU binding. Note that the individual distances between the metal-binding residue Asp258 and Znα in subunits B and D are different.

**Figure 6 ijms-22-06861-f006:**
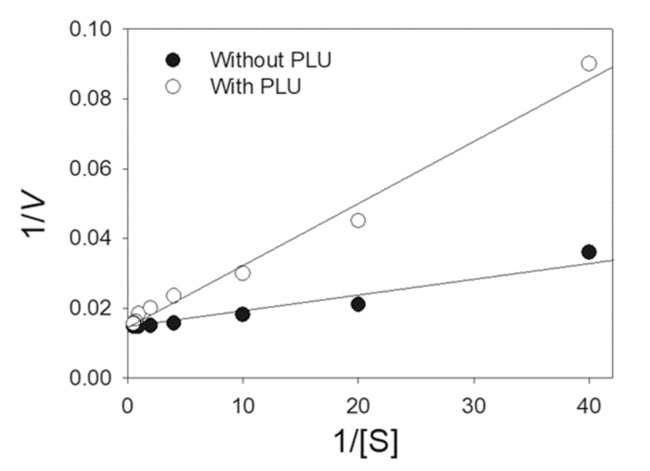
PLU is a competitive inhibitor of ScDHOase. PLU (50 μM) was included in the standard assay for analyzing the enzyme activity under different substrate concentrations. Inhibition of ScDHOase by PLU resulted in a Lineweaver–Burk plot where the lines cross the y-axis at a similar point, indicating that PLU is a competitive inhibitor for ScDHOase. Data points are an average of three determinations within the 10% error.

**Figure 7 ijms-22-06861-f007:**
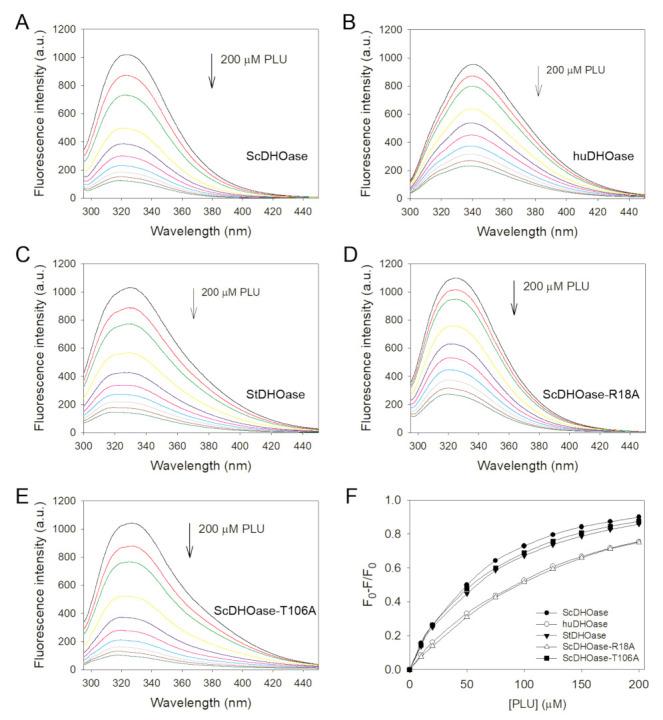
The data used to calculate the *K*_d_ values of various DHOases. (**A**) The fluorescence emission spectra of ScDHOase with PLU of different concentrations (0–200 μM; 0, 10, 20, 50, 75, 100, 125, 150, 175, and 200 μM). The decrease in intrinsic fluorescence of protein was measured at 324 nm upon excitation at 280 nm with a spectrofluorometer. The fluorescence intensity emission spectra of ScDHOase significantly quenched with PLU. (**B**) The fluorescence emission spectra of huDHOase with PLU of different concentrations (0–200 μM). (**C**) The fluorescence emission spectra of StDHOase with PLU of different concentrations. (**D**) The fluorescence emission spectra of ScDHOase–R18A with PLU of different concentrations. (**E**) The fluorescence emission spectra of ScDHOase–T106A with PLU of different concentrations. The huDHOase, StDHOase, ScDHOase–R18A, and ScDHOase–T106A individually displayed strong intrinsic fluorescence with peak wavelengths of 340, 330.5, 324.5, and 328 nm when excited at 280 nm. (**F**) An aliquot amount of PLU was added to the enzyme solution to determine the *K*_d_. The *K*_d_ was obtained by the following equation: Δ*F* = Δ*F*_max_ − *K*_d_(Δ*F*/[PLU]). Data points are an average of 2–3 determinations within a 10% error.

**Figure 8 ijms-22-06861-f008:**
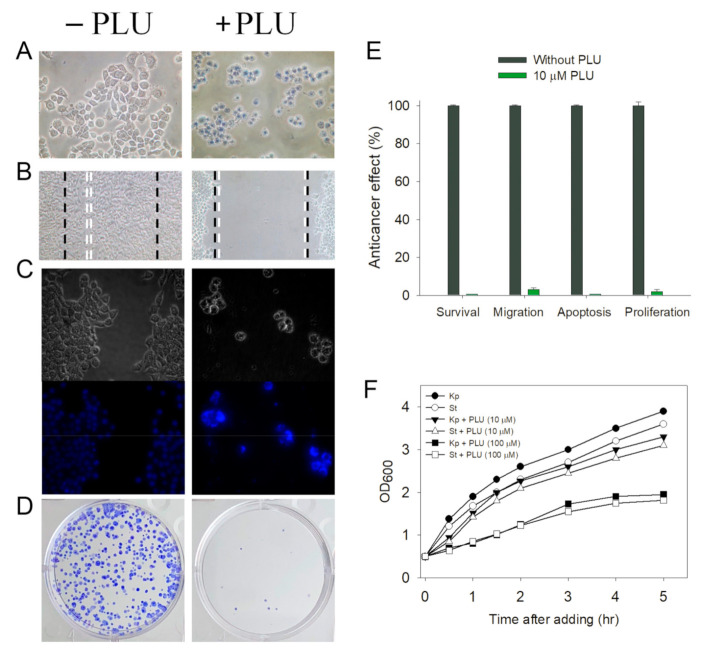
Anticancer and antibacterial activities of PLU. (**A**) The trypan blue assay—4T1 cells incubated with PLU (10 μM). After 24 h, the anticancer potentiality was estimated. (**B**) The wound-healing assay—4T1 cells were seeded in 24-well plates. Cells were incubated with PLU (10 μM) for 24 h to allow migration. (**C**) Hoechst staining assay—an amount of 10 μM PLU-induced apoptosis with DNA fragmentation was observed. (**D**) The clonogenic formation assay—pretreatment with 10 μM PLU significantly suppressed the proliferation and colony formation of 4T1 cells. (**E**) Effects of PLU on cell survival, migration and proliferation, and nuclear condensation—a concentration of 10 μM PLU was effective for anti-4T1 cancer cells. (**F**) PLU inhibited bacterial growth—*K. pneumoniae* and *S. enterica* serovar Typhimurium LT2 cells were grown to 0.5 OD_600_ at 37 °C, and then PLU was added into the medium. An amount of 100 μM PLU significantly inhibited the growth of these two pathogens. PLU may inhibit the pyrimidine biosynthesis, resulting in its antibacterial activities.

**Figure 9 ijms-22-06861-f009:**
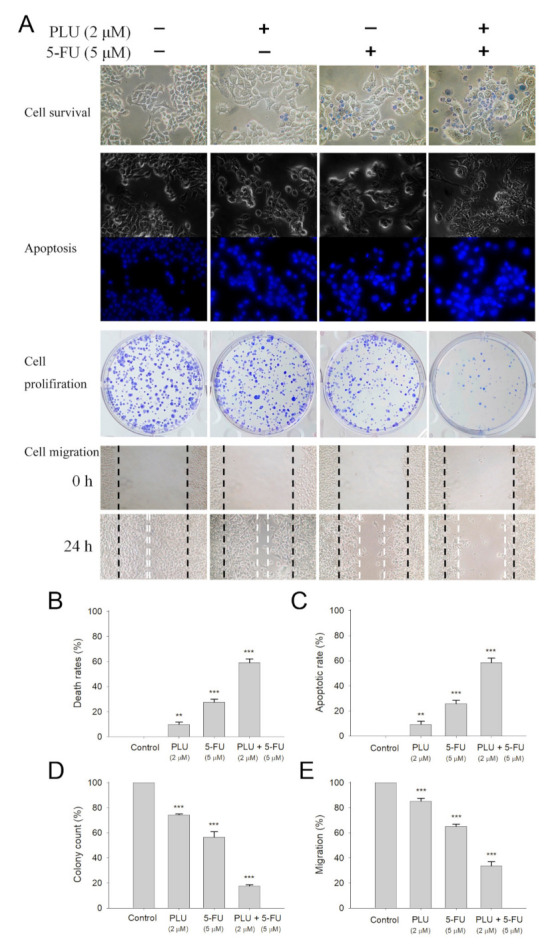
The synergistic anticancer effects of PLU and 5-FU. (**A**) Effects of PLU and 5-FU on cell survival, migration and proliferation, and apoptosis. (**B**) Trypan blue dye exclusion staining—4T1 cancer cells incubated with PLU and 5-FU. (**C**) Hoechst staining—PLU- and 5-FU-induced apoptosis with DNA fragmentation was observed in 4T1 cells. (**D**) Clonogenic formation assay—pretreatment with PLU and 5-FU significantly suppressed the proliferation and colony formation of 4T1 cells. (**E**) The wound-healing assay—PLU and 5-FU significantly inhibited cell migration. ** *p* < 0.01 and *** *p* < 0.001 compared with the control group.

**Figure 10 ijms-22-06861-f010:**
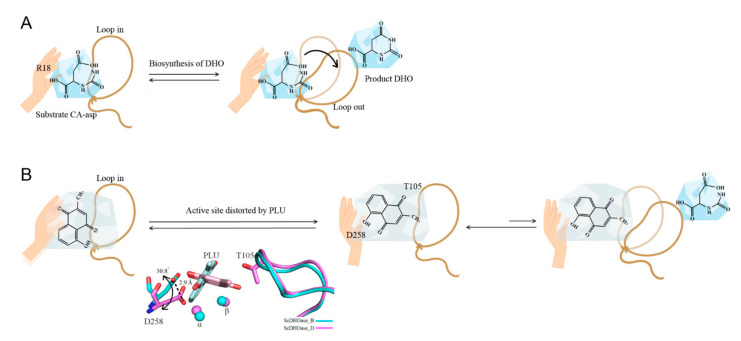
Proposed inhibition model. (**A**) The dynamic loop for the substrate entrance of DHOase. The structure of EcDHOase reveals a flexible loop that extends toward the active site when CA-asp is bound (the loop-in mode) or moves away from the active site, facilitating the release of the product DHO (the loop-out mode). The flexible loop in EcDHOase is crucial for stabilizing the transition state and catalysis, supporting that the movement of this loop is a part of the catalytic cycle. (**B**) The active site distorted by PLU. Analysis of 5-FU- and 5-AU-complexed structures of ScDHOase indicated loop-in binding mode; however, they were not inhibitors.

**Table 1 ijms-22-06861-t001:** Data collection and refinement statistics.

Data collection	
Crystal	PLU–ScDHOase
Wavelength (Å)	0.9
Resolution (Å)	45.9–3.60
Space group	*P*2_1_
Cell dimension *a*, *b*, *c* (Å) *β* (°)	85.24, 88.36, 103.42 95.16
Redundancy	1.2 (1.2)
Completeness (%)	94.54 (83.76)
<I/σ_I_>	5.4 (1.3)
CC_1/2_	0.942 (0.78)
Refinement	
Resolution (Å)	45.9–3.60
No. reflections	16975
*R*_work_/*R*_free_	0.26/0.31
No. atoms	
Ligands	54
Protein	1460
Zinc	8
Water	0
r.m.s deviations	
Bond lengths (Å)	0.002
Bond angles (°)	0.54
Ramachandran plot	
Favored (%)	93.94
Allowed (%)	5.71
Outliers (%)	0.35
PDB entry	7CA1

Values in parentheses are for the highest resolution shell. CC_1/2_ is the percentage of correlation between intensities of random half-data sets.

**Table 2 ijms-22-06861-t002:** Mutational analysis of ScDHOase.

ScDHOase	Specific Activity (μmol/mg/min)	Relative Activity (100%)	Corresponding Residue in CAD
Wild type	68 ± 4	100	huDHOase
H14A	<10^−3^	0	H1471 (metal binding)
H16A	<10^−3^	0	H1473 (metal binding)
R18A	<10^−3^	0	R1475 (substrate binding)
N43A	<10^−3^	0	N1505 (substrate binding)
K98A	<10^−3^	0	K1556 (metal binding)
T105A	0.57 ± 0.04	0.8	T1562 (dynamic loop)
T106A	0.15 ± 0.01	0.2	F1563 (dynamic loop)
H137A	<10^−3^	0	H1590 (metal binding)
H180A	<10^−3^	0	H1614 (metal binding)
D258A	<10^−3^	0	D1686 (metal binding and catalysis)
D258E	0.015 ± 0.003	0.02	D1686 (metal binding and catalysis)
H262A	<10^−3^	0	H1690 (substrate binding)

**Table 3 ijms-22-06861-t003:** Binding parameters of DHOases.

DHOase	λ_max_ (nm)	λ_em_ Shift (nm)	Quenching (%)	*K*_d_ Value (μM)
ScDHOase	from 324 to 317.5	6.5	87.7	64.8 ± 1.6
ScDHOase–R18A	from 324.5 to 320.5	4.0	75.1	181.0 ± 5.9
ScDHOase–T106A	from 328 to 321.5	6.5	90.0	70.8 ± 2.6
huDHOase	from 340 to 338.5	1.5	75.6	150.9 ± 4.1
StDHOase	from 330.5 to 319.5	11.0	85.9	80.3 ± 2.5

## Data Availability

Atomic coordinates and related structure factors were deposited in the PDB with accession code 7CA1.

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
