# Peer review of "Plumbagin, a Natural Product with Potent Anticancer Activities, Binds to and Inhibits Dihydroorotase, a Key Enzyme in Pyrimidine Biosynthesis"

_ijms, 2021, doi:10.3390/ijms22136861_

Round 1

Reviewer 1 Report

This manuscript describes endeavours to discover and characterise natural product-based inhibitors of DHOase as potential anticancer agents.  Initially, a study of substrate analogues were tested and found to be poorly inhibitory.  Next, screening of plant extracts led to discovery of plumbagin (PLU) as a DHOase inhibitor.  Inhibition is detected in several species’ DHOases.  PLU is stated to be a competitive inhibitor (although the data don’t seem to clearly show competitive inhibition or indeed what is outcompeted – DHO?, so this needs further evidence).  Co-crystallisation of PLU with ScDHOase led to definition of the binding mode, pose and interactions in the solid state.  Four monomeric complexes were seen in the asymmetric crystal unit: two bound to malate and two to PLU.  Similar binding modes and bonding interactions between protein and ligand were seen in each, although all were distinct, including the relative positions of the Zn atoms.  A flexible loop is seen to have very uniform position and conformation across all the monomeric subunits.

Site-directed mutagenesis was then conducted and all mutations of active site residues had a significant deleterious effect on catalytic activity of the DHOase.

Binding of several substrate analogues were studied through fluoresence quenching.  This study seems to be linked with the first results described and therefore this section would seem to be more sensibly positioned earlier in the manuscript, or at least linked better in the first paragraph of this section.

Clear activity against cancer cells by PLU was observed through cell proliferation, migration and apoptosis assays.  Some activity against bacteria was seen at relatively high concentration (100 uM). Synergistic effects on these measures by PLU with established DHOase binder and anticancer compound 5-FU were evident.

This work shows PLU binding and inhibiting DHOase, and a number of interesting conclusions have been drawn from this study about its mode of action and structural aspects of the binding (as covered in the discussion section).  This study is worthy of publication, although the manuscript should be cleaned up during the revision process.  A number of specific points to be addressed are described below.

Revisions:

CAD is mentioned in line 46 but not defined until line 50-52.

Structures of CA-asp, DHO and PLU shoudl be included in the Intro section.

Line 63: the benefits of natural products here are very limited – they are more than adjuvants and their diverse structures, in addition to biological activity.

Line 66-67: this is the wrong way round: 60% of anticancer drugs on the market are derived from or inspired by natural products.

Lines 67-71: the original literature on the first isolation of plumbagin, the anticancer results and the mode of action studies need to be presented here, not merely the review 26, based on the importance of the earlier work to the present study.

Line 73-74: the link between cytotoxicity and the quinone ring structure needs to be more clearly made here.  Is the quinone structure really unique?  This is surely a reasonably common motif in natural products.

Line 75-76: the need for a crystal structure of complexed PLU needs better justification here. 

Line 79: should read “...PLU is found to be a novel and...”

Line 81: “determine the binding interactions” might describe the results better than “where the binding occurred” as currently written.

Line 108: for better comparison with the Plasmodium DHOase results, inhibition (or lack thereof) in the nanomolar concentration range should be also quoted here.

Lines 90-117: the inhibition results should be graphed, or presented in a table or in Figure 1 with the structures, ideally as growth inhibition at a constant concentration, or as an IC50.

Line 125: should read “..screened for new DHOase...”

Figure 2B: the m/z (or, ideally, the mass spectrum) of the most abundant GC fraction shoudl be included here.  Also, the match probability with the data from the NIST and Wiley databases.

Line 140: define Sc, St, Kc etc.

Lines 150-155: this part about the co-crystallisation study doesn’t fit well here, as this section is about a competitive inhibition study, but is presumably needed to define why scDHOase was continued as the model.  Is it possible to change the order of this section and the next?

Line 165 and figure 3 caption line 175: should read “inhibition of DHOase...”

Line 166: the comment about inhibition of DHPase here and in the figure 3 caption is not well framed.  It appears to have been a result of a previous study (ref 35).  This needs to be clearly stated in the introduction and mentioned again here.

Line 156-176: the competitive inhibition assay needs to be more clearly described.  Only kinetic data for the interaction of PLU with DHOase is mentioned, yet PLU was stated to be a competitive inhibitor. Was displacement of the natural substrate DHO by PLU actually detected? 

Line 172: Figure 3B doesn’t show the signalling pathways (as the caption states), only words decribing the signalling outcomes.  Fix the wording.

Figure 4: A larger version of the tetrameric structure should be presented in the supporting information

Figure 4: Can the position of the flexible loop shown in Fig 4B be pointed out in the full protein structure in Fig 4A?  In Fig 4B, what is the source of the different superimpositions?  This should be clearly described in the caption.

Line 227: “are ciritical as those in...” doesn’t make sense.  Was it intended for this to describe the conserved nature of these residues?

Line 238: D258E doesn’t mean “Asp258 could replace E” but a mutation of Asp258 by a Glu.

Line 248: wording should be fixed, as the 5-FU binding to DHOase is not part of the present study.

Figure 6 is erroneously labelled – it doesn’t show Kd values, but instead the data used to calculate Kd.

Fig 7F: the 0 uM concentration trial shouldn’t be described as +PLU.

Line 380: the title DHOase is a novel target by  PLU is not clear.  Please reword.

Line 537: the actual solvents used for the extractions and their order should be included here or in the supporting information.

Reviewer 2 Report

In this study, Guan et al, report that the natural product plumbagin (PLU) binds and inhibits dihydroorotase (DHOase), an enzyme participating in the synthesis de novo of pyrimidine nucleotides. The authors also describe the binding mode of PLU to yeast DHO by solving the crystal structure of the complex. In addition, the authors report other results that are not well integrated and dilute the main message. These include, the inhibition of human DHOase by compounds other than PLU, an alanine scanning of active site residues in yeast DHO, and an anticancer activity test with tumoral 4T1 cells using PLU.

I think that the results are interesting but that a major revision is needed to highlight the importance and significance of the findings. The manuscript would improve if the authors did an effort to better integrate the results, perhaps eliminating those not needed, to endorse the central idea. 

Section "2.1. Inhibition of DHOase by using substrate analogs"
- Is this section needed to support the main finding of the study?
- If important, why not showing in a figure the actual activity/inhibition assays?
- Why only testing the effect of the compounds in human DHOase? What about the effect on the DHOase from yeast and bacteria that are used in the following results section?

Section 2.2, "Inhibition of DHOase by plant extracts"
- This section starts with the description of the isolation of PLU from plant extracts, which seems nearly identical to the results reported in a previous study from the same group:
Yen-Hua Huang et al. Identification and characterization of dihydropyrimidinase inhibited by plumbagin isolated from Nepenthes miranda extract. DOI: 10.1016/j.biochi.2020.03.005
Indeed, Figure 2 in the current manuscript shows the same information as Figure 1 in the already published article. If PLU isolation was done in the previous work it must not be presented in this new manuscript as a new result. 

- Why not showing the actual inhibition curves of the different DHOases by PLU? This is a central point of the article and perhaps deserves its own figure or table.

Section 2. 3. 
- Why not merging this section with section 2.2?
- The last part of this section and Figure 3B are not results, but an interpretation that should be moved to the discussion.

Section 2.4. Crystal structure of yeast DHO with PLU
- I miss in the manuscript some comment about the limitations of the electron density maps at 3.6 Å resolution, particularly for the interpretation of the density at the active sites.
- Figure 5 shows the electron density map corresponding to PLU bound to the active site. But this representation is misleading, since it appears that there is a clear extra density in the active site that allows a precise positioning of the PLU molecule. The text does not say the opposite, and the accompanying figures plot precise interactions and distances. If the authors want to show the quality of the electron density map, they should show the map for a larger section of the active site, and not only for the compound.

- I downloaded the model and structure factors from the PDB and recognize the effort of the authors to construct the best possible model. However, I find that the limited resolution does not allow a precise modeling of the ligands (malate or PLU) and that there is a large uncertainty in the binding mode of these compounds. Thus, I find that the claims in the manuscript about interaction details or active site deformations distances are not sufficiently sustained by the low-resolution data. 

Section 2.5: Mutational analysis of residues within the active site
- Why doing an alanine scanning to prove the conservation of key active site elements in yeast DHO? I find the results interesting, but this section is not linked to the main message of the manuscript. The authors should consider publishing these results as a different study.

Section 2.6: Binding specifities of yeast DHOase.
- In this section, the only relevant data is the Kd value for PLU and perhaps, how it compares with the values for other inhibitory compounds. I find this section confusing and distracting.

2.7 Structure-based binding analysis
- In this section, 2 active site elements that are key for the binding of the substrates reveal as not important for the binding of PLU: eliminating T106 in the flexible loop does not change the Kd for PLU, whereas mutating R18 has a modest effect (3-fold increase). I find the result very intriguing, but since the flexible loop is not important for PLU binding, why do the authors engage in the description of the movement of this flexible loop throughout the manuscript (including Fig. 4B, and Fig. 9)?

2.8. Anticancer activity of PLU against 4T1 cells.
- The authors refer to other articles showing that PLU is a potent anticancer agent. Then, why is important to show that PLU has an effect on 4T1 cells? Are these cells particularly sensitive to DHOase inhibition and pyrimidine supply? If not, although interesting, these data might not be relevant for the manuscript and could be removed?

- These experiments do not prove that the damaging effect of PLU on 4T1 cells is directly related DHOase inhibition. To prove it, perhaps one could test the effect of PLU in presence of exogenous uridine in the medium. In that case, cells could make pyrimidines through salvage pathways and bypass the PLU-inhibition of DHOase. 

Discussion
- I think that it cannot be concluded that "PLU can regulate cellular nucleotide synthesis to suppress cancer cell growth and survival (Figure 7) through inhibition of DHOase (Figure 3)."
As the author state, PLU inhibits multiple cancer signaling pathways, and thus, the results of Figure 7 do not clearly demonstrate that the effect is on pyrimidine synthesis and not in other cellular processes.

- I find that the discussion of the loop-in binding mode of PLU is not relevant, since the authors prove that the loop does not affect the affinity for the compound. 

- In "Section 3.5 Active site distorted by PLU", the authors should discuss the limitations imposed by the low resolution in the interpretation of the electron density maps. The claimed differences induced by the binding of PLU to the active site are not clearly seen in the downloaded structure and density maps.

- Throughout the paper there is confusing idea of "compounds that bind to the active site but are not inhibitors", for example 5-FU. To me, this is a misleading concept. If a compound binds at the active site with an affinity similar to that of the substrate, it will be a competitive inhibitor. If the affinity is lower, it will be easily displaced by the substrate, unless the compound is at high concentrations. Thus, all the compounds binding to the DHOase active site will inhibit at certain concentration. It does not matter if this compound can be hydrolyzed or not by DHOase. Even if it is hydrolysable, the compound inhibits by preventing the binding of the real substrate, DHO. Please consider revising the article to rephrase these ideas.

Methods
- The crystallization in presence of PLU are not sufficiently described, and would not allow to reproduce the experiment. Please provide more detail: concentration of PLU during crystal soaking, incubation time, cryoprotecting conditions... 

Additional comments

- In the introduction, when mentioning that yeast have a CAD-like with inactive DHOase domain, one should also add that yeast have a monofunctional DHOase encoded by an independent gene, which is the protein that is crystallized in the study.

- In the introduction: DHOases of type I are different to type II and type III in that they do not have a flexible loop. In these enzymes, the loop is short, and I am not sure if it is proven that is flexible. In any case, the loop does not fulfill the same role as the flexible loop in other DHOases. Apparently, it is the interaction with ATCase, rather than the flexible loop, what helps in the correct orientation of the substrate in the active site. Nonetheless, based on the results, the loop-in conformation appears to be irrelevant for PLU binding and a detailed description of the flexible loop might be irrelevant.

- Also, in relation to the flexible loop, reference 22 is not the correct one for describing the movement of the flexible loop. In the structure reported in that manuscript, Thoden et al. could not trace the loop because it was flexibly disordered. As far as I know, the description of the loop-in and loop-out conformations for the loop was in reference 31.

- I do not think that the 3.6 Åresolution allows for a detail description of the interaction between the protein an PLU. Thus, the claims of the "PLU interactome" seem unjustified.

I have appended a pdf with some additional corrections/suggestions. 

Round 2

Reviewer 2 Report

The authors have answered most of my queries and I do not have further comments on the article.